# Malaria infection and its association with socio-demographics, long lasting insecticide nets usage and hematological parameters among adolescent patients in rural Southwestern Nigeria

**Azeez Oyemomi IBRAHIM** [1] *, **Tosin Anthony Agbesanwa**[2], **Shuaib Kayode AREMU**[3], **Ibrahim Sebutu BELLO**[4,5], **Olayide Toyin ELEGBEDE**[1], **Olusegun Emmanuel GABRIEL-ALAYODE**[1], **Oluwaserimi Adewumi AJETUNMOBI**[1], **Kayode Rasaq ADEWOYE**[6], **Temitope Moronkeji OLANREWAJU**[7], **Ebenezer Kayode ARIYIBI**[7], **Adetunji OMONIJO**[7], **Taofeek Adedayo SANNI**[6], **Ayodele Kamal ALABI**[8], **Kolawole OLUSUYI**[1]

1 Department of Family Medicine, Afe Babalola University, Ado-Ekiti, Ekiti State, Nigeria, 2 Department of Family Medicine, Ekiti State University, Ado-Ekiti, Nigeria, 3 Department of ENT, Afe Babalola University, Ado-Ekiti, Ekiti State, Nigeria, 4 Department of Family Medicine, Obafemi Awolowo University Teaching Hospital Complex, Ile-Ife, Osun State, Nigeria, 5 Osun State University, Oshogbo, Nigeria, 6 Department of Community Medicine, Afe Babalola University, Ado-Ekiti, Ekiti State, Nigeria, 7 Department of Family Medicine, Federal Teaching Hospital, Ido-Ekiti, Ekiti State, Nigeria, 8 Department of Community Medicine, Federal Teaching Hospital, Ido-Ekiti, Ekiti State, Nigeria

* ibrahimao@abuad.edu.ng

## Abstract

### Background

There is increasing evidence suggesting that adolescents are contributing to the populations at risk of malaria. This study determined the prevalence of malaria infection among the adolescents and examined the associated determinants considering socio-demographic, Long Lasting Insecticide Nets (LLINs) usage, and hematological factors in rural Southwestern Nigeria.

### Methods

A hospital-based cross-sectional study was conducted between July 2021 and September 2022 among 180 adolescents who were recruited at a tertiary health facility in rural Southwestern Nigeria. Interviewer administered questionnaire sought information on their socio-demographics and usage of LLINs. Venous blood samples were collected and processed for malaria parasite detection, ABO blood grouping, hemoglobin genotype, and packed cell volume. Data were analyzed using SPSS version 20. A p-value <0.05 was considered statistically significant.

**Data Availability Statement:** All relevant data are within the paper and its Supporting Information files.

**Funding:** The authors received no specific funding for this work.

**Competing interests:** The authors have declared that no competing interests exist.

## Results

The prevalence of malaria infection was 71.1% (95% CI: 68.2%-73.8%). Lack of formal education (AOR = 2.094; 95% CI: 1.288–3.403), being a rural residence (AOR = 4.821; 95% CI: 2.805–8.287), not using LLINs (AOR = 1.950; 95% CI: 1.525–2.505), genotype AA (AOR = 3.420; 95% CI: 1.003–11.657), genotype AS (AOR = 3.574; 95%CI: 1.040–12.277), rhesus positive (AOR = 1.815; 95% CI:1.121–2.939), and severe anemia (AOR = 1.533; 95% CI: 1.273–1.846) were significantly associated with malaria infection.

## Conclusion

The study revealed the prevalence of malaria infection among the adolescents in rural Southwestern Nigeria. There may be need to pay greater attention to adolescent populations for malaria intervention and control programs.

## Introduction

Malaria continues to be a significant global health challenge and a major public health issue in several countries, including Nigeria [1]. The disease is transmitted in humans from one person to another through the bites of infected female mosquitoes of the anopheles [2, 3]. There are five species of parasites belonging to the genus *Plasmodium*, that transmit malaria, of which *P. falciparum* is the most prevalent [2, 3]. Globally, there were 247 million cases of malaria and 619,000 deaths in 2021 [4]. However, approximately 80% of all deaths due to malaria were concentrated in just 15 countries mainly in the Africa region [2]. Despite frantic efforts and interventions targeted at its elimination, 48% of the world population remains exposed to the risk of malaria, a figure substantially higher than the 40% widely cited [4]. Malaria accounts for the highest number of hospitalizations and outpatient visits in Africa [5, 6].

The prevalence of malaria infection among the adolescent patients varies from place to place, even within the same country, and may be due to differences in socio-demographic, environmental and climatic factors [7, 8]. In Nigeria, malaria prevalence of 66.7% [9], 64.0% [10], and 58.0% [11] have been reported by previous studies. Socio-demographic factors such as age, gender, education, occupation and income, which may directly affect human exposure and treatment, and climatic factors such as temperature, humidity, and rainfall which may support rapid growth and development of mosquito vectors have been well reported in urban and peri-urban centers [8, 11]. In Africa, malaria transmission is comparatively higher among the rural settings than the urban areas, and may be due to the higher vector density, poor housing status, and poor drainage systems in rural settings [8, 12].

Hematological profile may contribute to the clinical presentation of malaria, but not transmission of malaria infection [7, 13]. These changes may be related to the individual's packed cell volume, ABO blood grouping, and hemoglobin electrophoresis pattern [7, 13]. Anemia has been known to be the usual sign of the parasitic infection in endemic malaria areas with *P. falciparum* being an important contributor to anemia in children [7, 14]. Anemia is a condition characterized by a decrease in the number of red blood cells (RBCs) or a lower than normal amount of hemoglobin for a person's age and gender. The World Health Organization defines anemia as a hemoglobin level of less than 12g/dL in non-pregnant women aged 15 and older, and less than 13g/dL in men aged 15 and older [15]. These values reflect a deficiency in the number of red blood cells or inadequate hemoglobin levels, both of which are defining

features of anemia. Resistance to malaria is characterized by the development of an immune response acquired by the host and also depends on innate features as protective factor against infections. Such innate features include ABO blood grouping, sickle cell trait (HbAS), and sickle cell disease (HbSS) [16]. The role of ABO blood group and the genetic make-up in adolescents susceptibility to malaria have not been fully studied [7, 13, 14]. Though, several studies on malaria infection with ABO blood grouping and hemoglobin genotype have been reported in urban areas of Nigeria, however, many of such studies involved pregnant women and the under five children which are considered the most vulnerable groups [7, 8, 13]. There is a growing body of evidence suggesting that school-aged children and adolescents are also at risk of malaria [13, 14]. Moreover, due to a decline in transmission and exposure in some areas, the peak age of clinical attacks of malaria is shifting from very young to older children and adolescents [13, 15].

In order to adapt malaria control strategies to changes in transmission patterns in rural settings, there is urgent need for data on prevalence of *P. falciparum* infection and its associated factors among adolescent age groups in rural Nigeria. Identifying key risk factors for malaria infection are vital for decision makers in malaria prevention and control programs. Therefore, this study determined the prevalence of malaria infection among adolescent age groups and examined the associated determinants considering socio-demographic, Long Lasting Insecticide Nets Usage (LLINs) and hematological factors in rural community of Ekiti State, Southwestern Nigeria.

## Materials and methods

### Study area/design/period

A hospital-based cross-sectional study was carried out between July 2021 and September 2022 at the Federal Teaching Hospital, Ido-Ekiti (FETHI), Southwestern Nigeria. Ido-Ekiti is a rural community in Ekiti State and has a total land area of 332km2 and a total population of 159,114 inhabitants, according to the most recent population census conducted in 2006 [17]. The annual population growth rate is 3.2%, with the population in 2019 estimated to be 225,305 inhabitants [17]. The study area, Ido-Ekiti, is situated in a tropical rainforest characterized by climatic and environmental conditions that promote the growth and transmission of malaria [18, 19]. Malaria transmission occurs throughout the year, with the primary causative agent being P. falciparum [18]. The people are mainly farmers and traders in the informal sector with a relatively small portion of the working population and retirees in the formal sector [17]. In the study area, malaria transmission is perennial during the raining season (April–October) with *P. falciparum* being the major causative agent [19]. The health facility offers secondary and tertiary care to the people of its catchment area and its neighborhood states. The hospital has three outreach branches which were accredited for postgraduate residency training in family medicine, pediatrics and other core clinical specialties. Recruitment procedure and collection of data were done by the trained resident doctors and medical officers at the family medicine, Igogo and Ado clinic, and adult emergency unit of FETHI.

### Study population

This comprised all adolescent populations who presented to the family medicine, Igogo and Ado clinics, and adult emergency unit of the hospital.

### Inclusion criteria

Adolescent patients aged 10–19 years bracket who presented with febrile illness (with an axillary temperature $\geq 37.5°C$) and included those who consented (aged 18 years and above) and those who assented by adults (below 18 years of age).

### Exclusion criteria

Patients who were too ill that required immediate attention or those with mental illness. Also, those on treatment for malaria or have just completed anti malaria within two weeks prior to the conduct of this study.

### Sample size determination

This was determined using Araoye formular [20].

n = $Z^2$ $\underline{P(1-P)}$ and nf = $\frac{n}{1+\frac{n}{N}}$ with a prevalence (P) of 84.2% [21] in a previous study on prevalence of malaria infection amongst students of a southwest Nigerian federal university at 5% margin of errors and 95% confidence interval. where n = minimum sample size when the population (N) of the participants in the study area was greater than 10,000 over a period of one year and nf is the minimum sample size when the population (N) of the participants was less than 10,000. From the year 2019 medical records of adolescent patients with malaria, N was 780 in 2020, Z = 1.96%.

$$n = Z^2\,P(1-P)/d^2 = (1.96)^2 \times 0.842\,(1.000 - 0.842)\,/\,0.0025 = 204.$$

$$\text{Since nf} = \frac{n}{1 + \frac{n}{N}} = 204\,/\,1 + (204\,/\,780)$$

$$= 204\,/\,1.416806 = 161.$$

An attrition of 10% was added to cater for drop-out or loss of data and the minimum sample size was increased to 180 which was used for this study.

### Sampling technique

Systematic random sampling technique was used in this study. The sampling frame is the total number of febrile adolescent individuals expected during the study period. Data from the records department for 2019 gave a sampling frame of 540 over a period of 14 (54 weeks) months. Dividing the sampling frame by the sample size (180) gave a sampling interval of three (3). On each screening day, the third registered respondent was selected by systematic random sampling technique, and subsequently every third respondent were selected and process was repeated on each clinic day throughout the study period until the sample size of 180 was attained. Recruited individuals had their record cards tagged to prevent re-enrolment. The recruitment of the subjects was done by three trained resident doctors who served as research assistants while the researcher did the collection of data.

### Ethical clearance, consideration and consent

The study was approved by the Ethics and Research Committee of federal teaching hospital, Ido-Ekiti (ERC/2021/06/23/601A). When seeking consent from the respondents who were 18 year and above or assent from the guidance/parents of the respondent below aged 18 years, the methods and objectives of the study were explained clearly to the respondents individually.

For those respondents that could not read or write, the questionnaire was translated from English language to their local language by an independent interpreter who served as their legal guardian while back translation to English language was done to maintain response consistency. The respondents were told that they were free to refuse or disengage participation at any time without losing any benefit of care or favor to those that participated. Thus, written informed consent either by appending signature or thumbprint was obtained from all adult respondents and guardians/parents on behalf of their children before starting the study. Confidentiality and privacy were ensured throughout the study. The study was at no cost to the respondents. The respondents with positive malaria parasite were treated with the standard medication according to the national malaria drug policy. The reporting of this study conforms to the strengthening the Reporting of Observational Studies in Epidemiology (STROBE) statement [22].

## Data collection instruments and procedure

The two instruments for data collection were the interviewer administered questionnaire and data collection form. The questionnaire sought information about the respondents' socio-demographic characteristics (such as age, gender, education, occupation, and location), ownership and the usage of Long Lasting Insecticide Nets (LLINs).

## Clinical parameters of the respondents

**Sample collection.** Samples of blood (about 5ml) were obtained intravenously with the assistance of hospital phlebotomist. The blood samples were transferred into an *ethylenediaminetetraacetic acid* (EDTA) bottle to prevent blood coagulation.

1. **Microscopy for malaria parasite**: Two blood films, one thin and one thick were made from the blood samples. The thick and thin smears were prepared on clean, dry microscope glass slides and were allowed to dry. The thin smear was fixed in methanol and both smears were stained with 2% *Giemsa BDH* Laboratory supplies; Poole *BH* 15 ITD England [21]. The slides were viewed under a microscope using oil immersion at 100x magnification. Staining of the slides and parasite counting were made independently by two microscopists with discrepancies resolved by a senior microscopist who ensured quality control. The films were viewed for the presence of parasite. A negative result would mean absence of parasites at 200 high power fields. Parasite density was quantified against 200 leucocytes on an assumed leucocyte count of 8000 per ul of blood [23]. The degree of parasite density was graded as mild, moderate, and severe when the counts were <1000 parasites/ul of blood, 1000-9999/ul of blood, and ≥ 10,000/ul of blood, respectively, following the method described elsewhere [9].

**Parasites/ul of blood** = No. of asexual stages × 8000 leukocytes/200 leukocytes

2. **Determination of genotype**: The genotype of each respondent was determined as described by Ochei and Kolhatkar [24]. About 1ml of blood samples were withdrawn from the *EDTA* bottle and centrifuged at 2500 *rpm* for 5 minutes. The supernatant was then discarded and packed cells were washed with normal saline for three times. The red cells were lysed by adding an equal volume of distilled water, one quarter (1/4) of toluene followed by a drop of 3% *KCN* after the final wash. It was then mixed properly. Haemoglobin genotype separation was carried out using electrophoresis method as described by Cheesbrough [25].

3. **Determination of ABO blood group**. A drop of blood taking from the blood samples was placed on a clean slide in four concentric zones. Then, a drop of each of the *anti-sera*, *anti-*

*A*, *anti-B*, and *anti-D* was added and mixed with each of the blood samples with the aid of a sterile stick. Blood groups were determined on the basis of agglutination method [26].

4. **Determination of Packed cell volume (PCV)**: A micro-haematocrit tube was filled with blood and centrifuge in a micro-haematocrit rotor at 10,000 *rpm* for 5 minutes. PCV was read using the micro-haematocrit reader, and recorded as, no anemia (PCV≥30%), mild anemia (25–29%), moderate anemia (20–24%), and severe anemia (<20%) [25].

**Quality control.**   To ensure that the authorized standard operating procedure was followed for all the investigations, a senior microscopist was recruited to examine the slides for quality control. Our methodology may be reproduced by fellow researchers if they so desire.

**Treatment of the respondents.**   The results of the investigations were transmitted to the managing physicians through the patients. Respondents with malaria parasitaemia were treated with the standard medication in accordance with the national malaria drug policy.

**Data entry and analysis.**   Data collected were checked, cleaned and entered into EPI Info Version 7.0 and were exported to *IBM SPSS* for window version 21.0 (*IBM* Corp., Armonk, NY, USA), respectively for analysis. Quantitative data were expressed as mean ± standard deviation. Frequencies were used to determine the prevalence of malaria infection in the respondents. Binary logistic regression was employed to assess the determinants of *Plasmodium* infection. Variables significant at P-value < 0.05 in the univariate logistic regression were selected for multivariate logistic regression analysis model. Odds ratios with 95% confidence intervals were calculated. For each category of the independent variable, the one with least

**Table 1. Socio-demographic characteristics of respondents.**

| Variable | Frequency | Percentage |
|---|---|---|
| | (N = 180) | (%) |
| **Age (in years)** | | |
| 10–14 | 97 | 53.9 |
| 15–19 | 83 | 46.1 |
| *Mean age ± SD* | *14.5 ± 2.7* | |
| *Range* | *10–19* | |
| **Sex** | | |
| Male | 94 | 52.2 |
| Female | 86 | 47.8 |
| **Occupation** | | |
| Farmer | 16 | 8.9 |
| Trader | 18 | 10.0 |
| Student | 125 | 69.4 |
| Artisan | 21 | 11.7 |
| **Education** | | |
| None | 8 | 4.4 |
| Primary | 47 | 26.1 |
| Secondary | 101 | 56.1 |
| Tertiary | 24 | 13.4 |
| **Domicile** | | |
| Rural | 139 | 77.2 |
| Urban | 41 | 22.8 |

Olds ratio was categorized as constant reference "Ref" and P-value < 0.05 was considered to be statistically significant.

## Results

A total of 180 respondents were screened during the study period. The distribution according to socio-demographic characteristics showed that the majority of the respondents 97 /180 (53.9%) were young children within the ages of 10–14 years. The mean age of the respondents was 14.5±2.7 years (range: 10–19 years). There were more males 94/180 (52.7%) than females 86/180 (47.8%). Many of the respondents were unemployed 125 (69.4%) and majority were secondary school students 101 (56.1%), and were rural dwellers 139 (77.2%), (Table 1).

The majority of the respondents tested positive to only *Plasmodium falciparum* 128/180; 71.1% (95% CI: 68.2%-73.8%) which was identified from thin blood smear. Of the 128 diagnosed with *Plasmodium falciparum*, 72/128 (40.0%), 40/128(22.0%), and 16/128 (9.0%) had mild, moderate, and severe parasitaemia from thick blood smear respectively (Table 2).

After adjusting for possible confounders; the odds of being infected with *P. falciparum* infection was 2.094 times (95% CI:1.288–3.403) higher among the respondents who had no formal education, 4.821 times (95% CI: 2.805–8.287) higher among the respondents who were rural dwellers, 1.950 times (95% CI: 1.525–2.505) higher among the respondents who were not using LLINs, 3.420 times (95% CI: 1.003–11.657) higher among the respondents who were hemoglobin genotype A, 3.574 times (95% CI: 1.040–12.277) higher among the respondents who were genotype AS, 1.815 times (95% CI: 1.121–2.939) higher among the respondents who were rhesus positive, and 1.533 times (95% CI: 1.273–1.846) higher among the respondents who had severe anemia, (Table 3).

## Discussion

The study identified the prevalence of malaria infection and examined the socio-demographics, LLINs Usage and hematological parameters-based factors that determine the likelihood of malaria infection among adolescents aged 10–19 years in rural Southwestern Nigeria. The prevalence of malaria infection (71.1%) found in this study was comparable to a cross sectional study conducted among school aged participants in Abia Southeastern Nigeria which reported prevalence of 68.1% [27]. This may be due to the similarity in the study population, climatic factors, and the use of bed nets. In contrast, the prevalence of malaria in this study was high than 12.9% reported in a cross sectional study conducted in Kano Northwestern Nigeria [28]. It was also higher than 35.7% reported in another study conducted in Kaduna North-central Nigeria [29], and 36.6% reported in Plateaus North central Nigeria [27]. This could be due to the differences in the prevailing climatic conditions and environmental factors of our study area

**Table 2. Prevalence of malaria parasitaemia among the respondents.**

| Variables | Frequency | Percentage(%) |
|---|---|---|
| **Malaria parasitaemia by density (N = 128)** | | |
| Mild parasitaemia (<1000 parasites/ul of blood) | 72 | 56.3 |
| Moderate parasitaemia (1000–9999 parasites/ul of blood) | 40 | 31.2 |
| Severe parasitaemia (≥10,000 parasites/ul of blood) | 16 | 12.5 |
| **Malaria parasitaemia from thin blood smear** | | |
| Positive | 128 | 71.1 |
| Negative | 52 | 28.9 |
| 95% Confidence Interval | (68.2%– 73.8%) | |

**Table 3. Crude and adjusted odd ratios for the factors significantly associated with malaria parasitaemia (N = 180).**

| Variable | +ve % | -ve % | COR (95% CI) | p-value | AOR (95% CI) | p-value |
|---|---|---|---|---|---|---|
| **Age (in years)** | | | | | | |
| 10–14 | 70 (54.7) | 27(51.9) | 1.117 (0.586–2.132) | 0.736 | 1.033 (0.856–1.246) | 0.863 |
| 15–19 | 58(45.3) | 25(48.1) | 1.000 | | 1.000 | |
| **Sex** | | | | | | |
| Male | 70(54.7) | 24(46.2) | 1.408 (0.737–2.689) | 0.299 | 1.104 (0.914–1.333) | 0.326 |
| Female | 58(45.3) | 28(53.8) | 1.000 | | 1.000 | |
| **Occupation** | | | | | | |
| Farmers | 12(9.4) | 4(7.7) | 6.000 (1.407–25.590) | **0.012** | 2.250 (0.923–4.387) | 0.234 |
| Traders | 11(8.6) | 7(13.5) | 3.143 (0.846–11.671) | 0.083 | 1.833 (0.903–3.723) | 0.158 |
| Students | 98(76.5) | 27(51.9) | 7.259 (2.664–19.779) | **<0.001** | 2.352 (0.863–4.337) | 0.234 |
| Artisan (ref) | 7(5.5) | 14(26.9) | 1.000 | | 1.000 | |
| **Education** | | | | | | |
| None | 6(4.7) | 2(3.9) | 4.200 (0.698–25.265) | 0.102 | 2.094 (1.288–3.403) | **0.013** |
| Primary | 71(55.5) | 30(57.7) | 9.567 (2.940–31.135) | **<0.001** | 1.800 (0.969–3.345) | 0.108 |
| Secondary | 41(32.0) | 6(11.5) | 3.133 (1.325–8.288) | **0.008** | 1.687 (0.876–2.754) | 0.127 |
| Tertiary (ref) | 10(7.8) | 14(26.9) | | 1.000 | 1.000 | |
| **Domicile** | | | | | | |
| Rural | 117(91.4) | 22(42.3) | 25 (11.363–55.621) | **<0.001** | 4.821 (2.805–8.287) | **0.015** |
| Urban (ref) | 11(8.6) | 30(57.7) | 1.000 | | 1.000 | |
| **LLITNs** | | | | | | |
| Yes (ref) | 36(28.1) | 42(80.8) | 1.000 | | 1.000 | |
| No | 92(71.9) | 10(19.2) | 10.733 (4.871–23.650) | **<0.001** | 1.950 (1.525–2.505) | **0.003** |
| **Genotype** | | | | | | |
| AA | 95(74.2) | 30(57.6) | 11.083 (2.184–56.242) | **0.001** | 3.420 (1.003–11.657) | **0.002** |
| AS | 27(21.0) | 7(13.4) | 13.500 (2.282–79.880) | **0.001** | 3.574 (1.040–12.277) | **0.004** |
| AC | 2(1.6) | 2(3.8) | 1.167 (0.124–10.991) | 0.893 | 1.125 (0.203–6.239) | 1.000 |
| SC | 2(1.6) | 2(3.8) | 3.500 (0.284–43.163) | 0.317 | 2.250 (0.470–10.779) | 0.726 |
| SS (ref) | 2(1.6) | 7(13.4) | 1.000 | | 1.000 | |
| **Blood group** | | | | | | |
| O | 63(49.3) | 19(36.5) | 9.118 (2.62–31.958) | **<0.001** | 2.881 (0.734–6.725) | 0.098 |
| A | 33(25.8) | 9(17.3) | 4.776 (1.333–17.111) | **0.012** | 2.380 (0.832–5.648) | 0.177 |
| B | 28(21.9) | 13(25.0) | 5.923 (1.582–22.172) | **0.005** | 2.561 (0.927–6.081) | 0.141 |
| AB (ref) | 4(3.1) | 11(21.2) | 1.000 | | 1.000 | |
| **Rhesus factor** | | | | | | |
| Rh D Positive | 118(92.2) | 38(73.1) | 4.347 (1.785–10.587) | **<0.001** | 1.815 (1.121–2.939) | **0.001** |
| Rh D Negative (ref) | 10(7.8) | 14(26.9) | 1.000 | | 1.000 | |
| **PCV** | | | | | | |
| ≥ 30% (ref) | 63(49.2) | 41(78.9) | 1.000 | | 1.000 | |
| 25–29% | 22(17.2) | 5(9.6) | 2.864 (1.004–8.164) | **0.043** | 1.345 (0.657–1.706) | 0.453 |
| 20–24% | 17(13.3) | 4(7.7) | 2.766 (0.869–8.806) | 0.076 | 1.336 (0.453–2.748) | 0.127 |
| < 20% | 26(20.3) | 2(3.8) | 8.460 (1.905–37.579) | **0.001** | 1.533 (1.273–1.846) | **0.001** |

*ref–reference category COR–Crude Odd Ratio AOR–Adjusted Odd Ratio +ve–positive parasitaemia, -ve—negative parasitaemia*

compared to other studies that were conducted in Northern Nigeria. Previous study across the regions of Nigeria had reported a higher prevalence of malaria infection in the Southwestern Nigeria compared to the Northern Nigeria [27, 30]. Thus, differences in settings, season

variations, and environment factors conducive for malaria vectors to thrive might be responsible for this finding [27, 30]. Furthermore, the prevalence of malaria in this study was lower than the results of a cross sectional study conducted among students of a southwest Nigerian federal university (80.6%) [31], and in Southern Tanzania (78.0%) [32]. The low prevalence in this study compared to the other previous malaria study in Akwa, Southeastern Nigeria might be attributed to the usage of insecticide treated nets as well as the setting. However, it is noteworthy to state that malaria prevalence recorded in this study is of public health importance and could be attributed to several factors in the study area such as climatic, vegetation and environmental factors and the increase chances of participants contact with malaria [30].

Furthermore, our finding revealed that *P. falciparum* was the only *Plasmodium* species encountered in this study. This is comparable to cross sectional study conducted in other malaria endemic setting [14]. This finding was not surprised given the fact that *P. falciparum* is the dominant *Plasmodium* species in Southwestern Nigeria [30, 33], Africa [34], and 99.7% estimated malaria cases of *P. falciparum* have been documented from WHO African regions [1]. It is responsible for the reported high morbidity and mortality among children, especially in sub-Sahara Africa countries. Using multivariate regression for factors that were significantly associated with malaria infection in this study, respondents who had no formal education were 3.403 times more likely to increase the odds of malaria infection as compared to those who had formal education. This was similar to cross sectional studies conducted in Ibadan Southwestern Nigeria [11], Kano Northwestern Nigeria [9], and Kenya [35], which showed that level of education significantly influences the knowledge, attitude, and practice of people in various malaria interventions, treatment, and control [9, 11, 33, 35].

Similarly, the respondents who were rural dwellers were 4.821 times more likely to increase the odds of malaria infection. This was in agreement with studies conducted in Ibadan Southwestern Nigeria [11] and Kenya [36]. However, other cross sectional studies conducted in Kano Northwestern Nigeria [28], and other Africa country [37], have reported a high prevalence of malaria infection among the urban residents. The increased odds of malaria infection among the rural dwellers in this study may be due to their low level of education, environmental factors, and improper use of LLINs which have been identified as contributory factors for malaria infections in this study. Other studies have attributed the increased odds of malaria infection in rural areas to lack of social mobilization concerning malaria prevention [35, 36].

In this study, less than half of the respondents (43.3%) reported usage of LLINs, despite that the majority are in possession of LLINs. This was in agreement with cross sectional studies conducted in various parts of Africa, where LLIN has been shown to be at its lowest among school children [7, 11, 27]. Awareness campaign on malaria prevention with sustained and effective communication strategies geared towards transmission of ownership into usage is recommended. The use of LLINs in this study was found to be significantly associated with reduction of malaria infection as the results showed that respondents who denied usage of LLINs were 1.950 times more likely to have malaria infection as compared to the respondents who did. This finding was consistent with the reports of studies conducted in central Nigeria [7], Southwestern Nigeria [11], and western Kenya [38]. Appropriate utilization of LLINs is one of the major cost effective interventions adopted for the prevention of malaria transmission in endemic settings [30, 33]. This finding further reinforces the usefulness of LLINs and calls for its continuous awareness campaign to improve its usage in rural settings [30].

The study showed that respondents who were of hemoglobin genotype *AA* were associated with increased odds of malaria infections. This findings was consistent with findings in other studies reported in literatures [36, 37, 39, 40]. From results obtained in this study and reports of previous and similar studies, it is also clear that malaria infection rates were lowest for *HbSS* [38, 41]. The high prevalence of malaria in the *AA* hemoglobin genotype variant compared to

*SS* may be due to high rate of oxygen consumption and a large amount of hemoglobin ingested in the peripheral blood during the stage of replication [39–41]. Thus, malaria parasite found hemoglobin *AA* more conducive to strive than hemoglobin genotype *SS* because the red cells are conducive for the growth and development of the parasites [42].

This study also showed that respondents who were rhesus positive were 1.815 times more likely to increase the odds of malaria infection as compared to respondents who were rhesus negative. The pathophysiology of rhesus blood group in the protective or risk factor for malaria infection remains unclear. However, it may be due to alteration of adhesive properties of the parasites to the red blood cells (RBCs) membrane and subsequent sequestration of RBCs in less desirable locations for *Plasmodium* infection [43]. There are very few studies that looked at the association between malaria and rhesus blood group. The finding in the current study was comparable to the study by Mukhtar et al, which found a positive association between malaria parasites and rhesus positive [44]. In contrast to the current study, Bamou and Sevid-zem found that rhesus factor has no impact on malaria infection [45]. Further studies using prospective designs and a definitive method of *Plasmodium* identification are needed to identify the association between *P.falciparum* and rhesus individuals.

In this study, anemia was found to be significantly associated with malaria infection. This finding was consistent with the reports of previous study conducted among school aged populations in Abia State [27]. However, study by Umaru and colleague in Kaduna State Nigeria found no significant association between anemia and malaria infection [29]. The mechanism through which malaria infection causes anemia in rural dwellers is multi-factorials. It is possible that the respondents may have been anemic initially as a result of micronutrients deficiency, and further attack of malaria may worsen the level of anemia in them [30]. The results of the current study demonstrate that adolescents with a reduction in packed cell volume are at a higher risk of malaria infection. Specifically, the study found an increased prevalence of malaria infection among this group, highlighting the association between low packed cell volume and malaria risk [30, 39]. Physiologically, school aged children are susceptible to malaria infection due to their loss of acquired immunity, and thus increased likelihood of being anemic [27, 34].

## Limitations

This study was based on cross sectional design and thus, had limited opportunities to measure any causal association between malaria infection and other factors. The lack of children less than 10 years old was also a possible limitation for having a real burden of malaria in these older children and adolescents. Hemoglobin level was only assessed at base line; measurement of hemoglobin level for each suspected case could have provided a better assessment of the association between anemia and malaria in the study area [38]. Despite these limitations, the study provided a vital information regarding the burden of, and associated risk factors for malaria infection among the adolescents, which could be needed to recommend appropriate interventions for malaria prevention and control in rural Southwestern Nigeria [30, 33].

## Conclusion

In this study, the prevalence of malaria infection was 71.0%. The respondents with lack of formal education, rural residence, not using LLINs, blood hemoglobin genotype AA and AS, rhesus positive individuals, and having severe anaemia were significantly associated with the risk of malaria infection. The study demonstrated the need to pay attention to adolescent populations in rural settings for malaria interventions and control programs. The results may assist the stakeholders in recommending appropriate interventions for malaria prevention and control in rural Southwestern Nigeria [28].

## Supporting information

**S1 File.**
(DOCX)

**S1 Data.**
(XLSX)

## Acknowledgments

The authors would like to appreciate the participants, and also nurses, resident doctors and medical officers at the family medicine, outreach centers, and adult emergency medicine departments and the management of FETHI where the study was conducted.

## Author Contributions

**Conceptualization:** Azeez Oyemomi IBRAHIM.

**Data curation:** Azeez Oyemomi IBRAHIM, Shuaib Kayode AREMU, Ibrahim Sebutu BELLO, Oluwaserimi Adewumi AJETUNMOBI, Kayode Rasaq ADEWOYE, Ebenezer Kayode ARIYIBI, Ayodele Kamal ALABI, Kolawole OLUSUYI.

**Formal analysis:** Azeez Oyemomi IBRAHIM, Shuaib Kayode AREMU.

**Funding acquisition:** Azeez Oyemomi IBRAHIM, Tosin Anthony Agbesanwa, Olusegun Emmanuel GABRIEL-ALAYODE, Oluwaserimi Adewumi AJETUNMOBI, Kayode Rasaq ADEWOYE, Taofeek Adedayo SANNI.

**Investigation:** Azeez Oyemomi IBRAHIM, Tosin Anthony Agbesanwa, Ibrahim Sebutu BELLO, Olayide Toyin ELEGBEDE, Olusegun Emmanuel GABRIEL-ALAYODE, Temitope Moronkeji OLANREWAJU, Adetunji OMONIJO, Ayodele Kamal ALABI.

**Methodology:** Azeez Oyemomi IBRAHIM, Ibrahim Sebutu BELLO, Taofeek Adedayo SANNI, Kolawole OLUSUYI.

**Project administration:** Azeez Oyemomi IBRAHIM, Oluwaserimi Adewumi AJETUNMOBI, Adetunji OMONIJO.

**Resources:** Azeez Oyemomi IBRAHIM, Olayide Toyin ELEGBEDE, Olusegun Emmanuel GABRIEL-ALAYODE, Adetunji OMONIJO, Ayodele Kamal ALABI, Kolawole OLUSUYI.

**Software:** Oluwaserimi Adewumi AJETUNMOBI, Temitope Moronkeji OLANREWAJU, Ebenezer Kayode ARIYIBI.

**Supervision:** Shuaib Kayode AREMU.

**Validation:** Shuaib Kayode AREMU, Olayide Toyin ELEGBEDE, Taofeek Adedayo SANNI.

**Visualization:** Shuaib Kayode AREMU.

**Writing – original draft:** Ebenezer Kayode ARIYIBI.

**Writing – review & editing:** Kayode Rasaq ADEWOYE, Kolawole OLUSUYI.

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
