## [Decision Letter · Decision Letter 0]

22 Feb 2023

PONE-D-23-02179MALARIA INFECTION AND ITS ASSOCIATION WITH SOCIO-DEMOGRAPHICS, LONG LASTING INSECTICIDE NETS USAGE AND HEMATOLOGICAL PARAMETERS AMONG ADOLESCENT PATIENTS IN RURAL SOUTHWESTERN NIGERIAPLOS ONE

Dear Dr. Ibrahim,

Thank you for submitting your manuscript to PLOS ONE. After careful consideration, we feel that the manuscript does not fully meet PLOS ONE’s publication criteria as it currently stands. Therefore, we invite you to submit a revised version of the manuscript that addresses the points raised during the review process. In addition to the comments raised by the reviewers, the authors will need to re-work the data analysis in Table 3, particularly the percentages and carefully go through the entire manuscript to correct grammatical errors as well as italicize all scientific names.

We look forward to receiving your revised manuscript.

Kind regards,

Segun Isaac OYEDEJI, Ph.D

Academic Editor

PLOS ONE

2. In the ethics statement in the Methods, you have specified that verbal consent was obtained. Please provide additional details regarding how this consent was documented and witnessed, and state whether this was approved by the IRB

https://pubmed.ncbi.nlm.nih.gov/36051785/

In your revision ensure you cite all your sources (including your own works), and quote or rephrase any duplicated text outside the methods section. Further consideration is dependent on these concerns being addressed.

“Self funded by authors”

“No competing interest”

Reviewers' comments:

Reviewer's Responses to Questions

**Comments to the Author**

1. Is the manuscript technically sound, and do the data support the conclusions?

Reviewer #1: Partly

Reviewer #2: Partly

Reviewer #3: Yes

2. Has the statistical analysis been performed appropriately and rigorously? 

Reviewer #1: Yes

Reviewer #2: No

Reviewer #3: Yes

3. Have the authors made all data underlying the findings in their manuscript fully available?

Reviewer #1: Yes

Reviewer #2: No

Reviewer #3: Yes

4. Is the manuscript presented in an intelligible fashion and written in standard English?

Reviewer #1: Yes

Reviewer #2: Yes

Reviewer #3: Yes

5. Review Comments to the Author

Reviewer #1: The study of malaria prevalence in adolescent children is a good idea because it was carried out in a relatively neglected population, when it comes to malaria research. Most malaria studies usually involve pregnant women and children below the age of 5 years. Since children and pregnant women are mostly targeted in malaria control measures, there is a risk of malaria epidemiology changing to adolescent and older population. Hence this study is has significant epidemiologic relevance to malaria control. The study was well designed and the sampling method looks very good.

The discussion needs a major review, for instance there was no basis for comparing the prevalence of this study to those whose patient selection was not similar. The result of this study cannot be compared to that of Akure or Lagos where the study populations were apparently health and of different ages.

Outdoor transmissions or development of immunity were not assessed in this study, so they cannot be implied as stated in the discussion. Same with exposure to constant mosquito bites in participants, which was assumed.

Authors should indicate if prevalence of malaria parasitaemia in table 2 was derived from thick film or thin film? Findings from thick film and thin film were not clearly stated in the result section.

Some few grammatical errors:- First line of quality control, “is” should be changed to “was”, Sides not slide, reproduced not reproduce

Regarding the treatment of respondents: Better to clarify that the results were “transmitted to the managing physicians through the patients” than to just write that test results were given to respondents

How was the P. falciparum specie determined in this study?

Limitations- Self report of too many limitations is not good, especially when some of them are avoidable.

Sample size should not be a limitation since it was scientifically derived

Failure to find out the quality of LLITNs is an admission of negligence by the research team, rather than limitation. Better to keep quiet about this.

The conclusion of high malaria prevalence in this study area cannot be safely derived from febrile subjects. This gives a impression that the subjects were apparently healthy.

Overall, this is a good study that is worthy of attention if the results and discussion are well revised

Reviewer #2: The authors provided data on the MALARIA INFECTION AND ITS ASSOCIATION WITH SOCIODEMOGRAPHICS, LONG LASTING INSECTICIDE NETS USAGE AND HEMATOLOGICAL PARAMETERS AMONG ADOLESCENT PATIENTS IN RURAL

SOUTHWESTERN NIGERIA. While I commend the efforts of the authors, I do not feel that the work should be accepted for publication in PlosONE journal following the reasons below:

1. Its difficult for me to trust the microscopy results. Since this is the only method used to confirm Malaria parasite, it would have been nice to follow a standard microscopy procedure. Two microscopists should have done the reading with the third microscopists confirming any discrepancies arising from the reading of the two initial microscopists. Unfortunately, this was not done and that spurs me to recommend rejection.

2. For the calculation of the odd ratio, I feel also that the control group is very small for any meaningful inference.

3. The limitations listed in the manuscripts do not look like an insurmountable challenges but shows that the work was not well planned.

5. I strongly recommend the work to other lower journals where the public could still be reached with the outcome of the research.

6. I am worried that the author did not provide the parasite density results of the microscopy, at least to see to what degree the variables could influence parasite density.

7. The author mentioned that this s a cross-sectional study but none of the results were presented to indicate it so.

8. The -ve and +ve in Table 3 is very confusing. The legend could not provide what they stood for.

Reviewer #3: The methods used and the results obtained have supported the conclusion. All findings are fully described in the manuscript and written in standard English.

Review:

Malaria remains a public health problem. The impacts of different interventions against malaria such as a decline in transmission and exposure in some areas have been recorded, and the peak age of clinical attacks of malaria is shifting from very young to older children and adolescent.

In order to adapt malaria control strategies to changes in transmission patterns, there is urgent need for data on prevalence of P. falciparum infection and its associated factors among adolescent age groups which are crucial for the effective implementation of preventive and health intervention programs. Thus, the authors have raised an important issue on determining the prevalence malaria infection among adolescent age groups and examine the associated determinants considering socio-demographic, Long Lasting Insecticide Nets Usage (LLINs) and hematological factors in rural community of Ekiti State, Southwestern Nigeria.

Minor comments

Introduction

The WHO epidemiological record and others references should be updated (for example, instead of WHO 2015 report use the actual report!)

Materials and methods

The determination of the genotype was not completely described. This should be done.

Results

- Table 2: the parenthesis should be completed;

- Table 3:

The wrong percentages for Domicile Urban (Ref) and genotype AC should be corrected;

What are the selection criteria of Ref. for each variable? This should be well explained in the methodology;

The age and sex should also be considered since the authors have divided the population into two groups (see Table 1)

Limitations:

The lack of children less than 10 years old is also a possible limitation for having a reel burden of malaria in these older children and adolescent.

References

In the section “References”: the ref. 6, 8, 9, 10, 11, 12, 14, 15, 23, 31, 32should be corrected.

6. PLOS authors have the option to publish the peer review history of their article (what does this mean?). If published, this will include your full peer review and any attached files.

Reviewer #1: **Yes: **Prof Efunshile Akinwale Michael

Reviewer #2: No

Reviewer #3: No

---

## [Author Response · Author response to Decision Letter 0]

3 Mar 2023

Summary of reviewer’s comments and Authors’ responses. (PONE-D-23-02179)

Reviewer’s comments Authors’ Response

Editor’s comment:

In addition to the comments raised by the reviewers, the authors will need to re-work the data analysis in Table 3, particularly the percentages and carefully go through the entire manuscript to correct grammatical errors as well as italicize all scientific names.

1. Please ensure that your manuscript meets PLOS ONE’s style requirements.

2. In the ethics statement in the Methods, you have specified that verbal consent was obtained. Please provide additional details regarding how this consent was documented and witnessed, and state whether this was approved by the IRB

3.We noticed you have some minor occurrence of overlapping text with the following previous publication(s), which needs to be addressed:

In your revision ensure you cite all your sources (including your own works), and quote or rephrase any duplicated text outside the methods section. Further consideration is dependent on these concerns being addressed.

“Self funded by authors”

a) Please clarify the sources of funding (financial or material support) for your study. List the grants or

organizations that supported your study, including funding received from your institution

.

“No competing interest”

Please complete your Competing Interests on the online submission form to state any Competing Interests. If you have no competing interests, please state "The authors have declared that no competing interests exist.", 

5. Review Comments to the Author

Reviewer #1

The discussion needs a major review, for instance there was no basis for comparing the prevalence of this study to those whose patient selection was not similar. The result of this study cannot be compared to that of Akure or Lagos where the study populations were apparently health and of different ages.

-Outdoor transmissions or development of immunity were not assessed in this study, so they cannot be implied as stated in the discussion. Same with exposure to constant mosquito bites in participants, which was assumed.

Authors should indicate if prevalence of malaria parasitaemia in table 2 was derived from thick film or thin film? Findings from thick film and thin film were not clearly stated in the result section.

Some few grammatical errors:- First line of quality control, “is” should be changed to “was”, Sides not slide, reproduced not reproduce

Regarding the treatment of respondents: Better to clarify that the results were “transmitted to the managing physicians through the patients” than to just write that test results were given to respondents 

How was the P. falciparum specie determined in this study?

Limitations- Self report of too many limitations is not good, especially when some of them are avoidable. Sample size should not be a limitation since it was scientifically derived

Failure to find out the quality of LLITNs is an admission of negligence by the research team, rather than limitation. Better to keep quiet about this.

The conclusion of high malaria prevalence in this study area cannot be safely derived from febrile subjects. This gives a impression that the subjects were apparently healthy.

Overall, this is a good study that is worthy of attention if the results and discussion are well revised

Reviewer #2: 

1. Its difficult for me to trust the microscopy results. Since this is the only method used to confirm Malaria parasite, it would have been nice to follow a standard microscopy procedure. Two microscopists should have done the reading with the third microscopists confirming any discrepancies arising from the reading of the two initial microscopists. Unfortunately, this was not done and that spurs me to recommend rejection.

2. For the calculation of the odd ratio, I feel also that the control group is very small for any meaningful inference.

3. The limitations listed in the manuscripts do not look like an insurmountable challenges but shows that the work was not well planned.

5. I strongly recommend the work to other lower journals where the public could still be reached with the outcome of the research.

6. I am worried that the author did not provide the parasite density results of the microscopy, at least to see to what degree the variables could influence parasite density.

7. The author mentioned that this is a cross-sectional study but none of the results were presented to indicate it so.

8. The -ve and +ve in Table 3 is very confusing. The legend could not provide what they stood for.

Reviewer #3: 

Minor comments

Introduction

The WHO epidemiological record and others references should be updated (for example, instead of WHO 2015 report use the actual report!)

Materials and methods

The determination of the genotype was not completely described. This should be done.

Results

- Table 2: the parenthesis should be completed;

ggg

- Table 3:

The wrong percentages for Domicile Urban (Ref) and genotype AC should be corrected;

What are the selection criteria of Ref. for each variable? This should be well explained in the methodology;

The age and sex should also be considered since the authors have divided the population into two groups (see Table 1)

Limitations:

The lack of children less than 10 years old is also a possible limitation for having a reel burden of malaria in these older children and adolescent.

References:

In the section “References”: the ref. 6, 8, 9, 10, 11, 12, 14, 15, 23, 31, 32should be corrected.

END

 Thank you for the efforts you put in to improve our manuscript. Your contributions are highly appreciated. Please find our responses below.

---The data has been reviewed and errors have been corrected. Check table 3 page 13. All scientific names have been italicized.

-The authors did not receive any funding for this work. This has been included in the cover letter.

1.The style requirement has been strictly followed.

2.Verbal consent was not specified in the manuscript. Rather, written informed consent was taken. Please see ethical consent and consideration under methodology in our first submission. The same was repeated in the revised manuscript on page 7, line 150 

3.Minor occurrence of overlapping with previous publications have been identified, and the statements have been modified. Appropriate references cited. See 

Page 14, line 285-287:Page 15 line 308-309: Page 16, line 329-332: page 18, line 369-372, 379-380

4.The authors received no specific funding for this work. 

This has been included in the cover letter.

5.The authors have declared that no competing interests exist. This has been included in the cover letter.

6.The data for this study will be made available as supporting information upon acceptance of this manuscript. This has been included in the cover letter.

Thank you for the efforts you put in to improve our manuscript. Your contributions are highly appreciated. Please find our responses below

The discussion has been reviewed. Comparisons of prevalence whose selections were not similar have been removed. See page 14, line 278-283, 

This statement “Outdoor transmissions or development of immunity” and “exposure to constant mosquito bites in participants” and their citations have been removed’ Page15, line 295

These comments have been addressed in table 2. See table 2. Page 12, line 251-253

Grammatical errors under methodology were noted and corrected. Page 10, line 215-217

This statement has been modified. See page 10, line 218-220 .

The determination of P.falciparum specie has been clearly explained on page 9, line 184—197

Some of the limitations have been removed. Page 18 line 364-372

The conclusion has been modified. Page 18, line 374-380

1.The missing information has been corrected on page 9,line 184-197

2.This was part of the limitations.

3.Limitations have been modified on page 18 line 364-372

6. Parasite density has been included in Table 2. Page 12, line 254-255

8. The legend has been provided in Table 3. Line 254-255.

The affected sentences in the introduction have been updated including the references. See page 3, line 58-63 and references 2-6 under references.

The description of genotype determination has been modified. See page 9, line 204-205

Table 2 has been modified page 12.

The percentages for Urban and genotype AC have been corrected. Check table 3, page 13

 For selection criteria of Ref, check page 10, line 228-229.

The age and sex have been included. Check table 3, on page 13

This statement has been added to the limitation. Check page 18, line 265-267.

All the quoted references have been corrected or replaced. See references.

END

---

## [Decision Letter · Decision Letter 1]

2 May 2023

PONE-D-23-02179R1MALARIA INFECTION AND ITS ASSOCIATION WITH SOCIO-DEMOGRAPHICS, LONG LASTING INSECTICIDE NETS USAGE AND HEMATOLOGICAL PARAMETERS AMONG ADOLESCENT PATIENTS IN RURAL SOUTHWESTERN NIGERIAPLOS ONE

Dear Dr. Ibrahim,

Thank you for submitting your manuscript to PLOS ONE. After careful consideration, we feel that it has merit but does not fully meet PLOS ONE’s publication criteria as it currently stands. Therefore, we invite you to submit a revised version of the manuscript that addresses the points raised during the review process.

There are some areas that need the attention of the authors in order to substantially improve the quality of the manuscript, particularly Tables 2 and 3, as well as some references. The authors may choose to request the services of a statistician to help with Tables 2 and 3.

In Table 2 for example, Malaria parasitaemia was classified as Mild parasitaemia, Moderate parasitaemia and Severe parasitaemia in 128 individuals who tested positive for P. falciparum. However, the frequency (N) was erroneously taken as the overall participants enrolled (180) rather than the total number positive for P. falciparum (128). Consequently, the percentages will not add up to 100%.

In Table 3, percentages were generally calculated across rows rather than across columns. Since you are comparing between positives and negatives, the percentages (as well as the statistical analyses) should be calculated/compared across columns rather than across rows. For example, a reader may wish to know the impact of using LLITNs on malaria by comparing the proportion of those using LLITNs among participants who tested positive, with the proportion of those using LLITNs among participants who tested negative.

In addition, under Domicile in Table 3, the Urban percentages (+ve and -ve) do not add up to 100%. Please check and correct.

We look forward to receiving your revised manuscript.

Kind regards,

Segun Isaac OYEDEJI, Ph.D

Academic Editor

PLOS ONE

Journal Requirements:

Additional Editor Comments:

INTRODUCTION

Page 3: Lines 57 and 58

Please remove italics from "genus"

Also correct the clause "... of which P. falciparum being the most prevalent." to "... of which P. falciparum is the most prevalent."

Page 3: Lines 57 and 58

Please review the statement "Hematological changes are other factors that may contribute to the transmission of microscopic malaria infection especially in children. Hematological profile may contribute to the clinical presentation of malaria, but not transmission of malaria infection. 

Page 4: Line 96

Please correct the clause "... determined the prevalence malaria infection..." to "... determined the prevalence of malaria infection...". 

DISCUSSION

Page 14: Line 278

Please correct "... higher..." to "...high..." since there was no other group for comparison.

Page 15: Line 297

Plasmodium should be uppercase letter. Please correct "... plasmodium species..." to "... Plasmodium species..." 

Reviewers' comments:

Reviewer's Responses to Questions

**Comments to the Author**

1. If the authors have adequately addressed your comments raised in a previous round of review and you feel that this manuscript is now acceptable for publication, you may indicate that here to bypass the “Comments to the Author” section, enter your conflict of interest statement in the “Confidential to Editor” section, and submit your "Accept" recommendation.

Reviewer #3: All comments have been addressed

Reviewer #4: (No Response)

2. Is the manuscript technically sound, and do the data support the conclusions?

Reviewer #3: Yes

Reviewer #4: Partly

3. Has the statistical analysis been performed appropriately and rigorously? 

Reviewer #3: Yes

Reviewer #4: Yes

4. Have the authors made all data underlying the findings in their manuscript fully available?

Reviewer #3: Yes

Reviewer #4: Yes

5. Is the manuscript presented in an intelligible fashion and written in standard English?

Reviewer #3: Yes

Reviewer #4: Yes

6. Review Comments to the Author

Reviewer #3: All small errors are highlighted in the document provided.

Results

Table 3: The wrong percentages for Domicile Urban (Ref) should be corrected;

References

In the section “References”:

- the ref. 3, 8, 17, 19, 24, 29, should be corrected.

- Line 453-455: font should be corrected.

Reviewer #4: Please, check my comments in the text especially the Discussion section which should be revisited as it does not appropriately address the content of your study

7. PLOS authors have the option to publish the peer review history of their article (what does this mean?). If published, this will include your full peer review and any attached files.

Reviewer #3: No

Reviewer #4: No

---

## [Author Response · Author response to Decision Letter 1]

16 May 2023

SUMMARY OF REVIEWERS’ COMMENTS AND AUTHOR RESPONSE (PONE-D-23-02179R1]

Author’s opening remark: I appreciate the academic editor and other reviewers at improving the quality of our work. Please find my responses to your comments and suggestions below.

Academic editor’s comment Author’s response

--In table 2 for example, Malaria parasitaemia was classified as mild, moderate and severe parasitaemia in 128 individuals who tested positive for P. falciparum. However, the frequency N was erroneously taken as the overall participants enrolled (180) rather than the total number positive for P.falciparum (128). Consequently, the percentages will not add up to 100%.

--In table 3, percentages were generally calculated across rows rather than across columns. Since you are comparing between positives and negatives, the percentages (as well as statistical analyses should be calculated /compared across column rather than across rows. For example, a reader may wish to know the impact of using LLITNs on malaria by comparing the proportion of those using LLITNs among participants who tested positive, with the proportion of those using LLITNs among participants who tested negative.

--In addition, under domicile in table 3, the urban percentages (+ve and –ve) do not add up to 100%. Please, check and correct.

Journal Requirements:

1. Please review your reference list to ensure that it is complete and correct. If you have cited papers that have been retracted, please include the rationale for doing so in the manuscript text, or remove these references and replace them with relevant current references. Any changes to the reference list should be mentioned in the rebuttal letter that accompanies your revised manuscripts. If you need to cite a retracted article, indicate the articles retracted status in the references list and also include a citation and full reference for the retraction notice.

2. Additional editor comments:

Introduction:

- Page 3: lines 57 and 58. Please remove italics from “genus”

- - Correct the clause” of which P.falciparum being the most prevalent “to –of which P. falciparum is the most prevalent”

- Page 3, line 57 and 58.

- Please review the statement “Hematological changes are other factors that may contribute to the transmission of microscopic malaria infection especially in children. Hematological profile may contribute to the clinical presentation of malaria, but not transmission of malaria infection.

-Page 4, line 96:

Please, correct the clause “determined the prevalence malaria infection ---“to determined the prevalence of malaria infection”

Discussion:

Page 14; LINE 278

Please correct “higher” to high” since there was no other group for comparison.

Page 15, line 297

-Plasmodium should be uppercase letter. Please correct “plasmodium species “to” Plasmodium species”

Reviewer#3: 

All small errors are highlighted in the document provided.

Results:

Table 3: The wrong percentages for domicile Urban (ref) should be corrected.

References:

In the section “References”, -the ref. 3,8,17, 24, 29 should be corrected.

-Line 453-455: font should be corrected.

Reviewer #4:

Please check my comments in the text especially the discussion section which should be revisited as it does not appropriately address the content of your study.

END# 

 Thank you for this observation. 

The correction has been effected in table 2, page 12, line 246

-This has been corrected in table 3, page13. All percentages were calculated/compared across columns.

-The correction has been effected in table 3, page 13.

-The reference list has been reviewed. The new references that replaced the previous references were highlighted in pink color under our revised manuscript with track changes. See the reference list 

This has been corrected on page 3, Line 53 under the introduction 

The correction effected on page 3, line 53-54

The statement has been reviewed. See page 3, line 70-72

-This statement has been modified on page 5, line 95

--This has been corrected on page 14, line 272 , line 

----This has been corrected on page 15, line 289

All small errors were corrected and highlighted in the manuscript. See the following pages;

Page 3, line 53,Page 4, line 84, Page 8,line 167, page 10 line 205, page 11 line 232, page 16, line 311, page 16, line 324

This has been corrected, See table 3, page 13

Some of the references (including 3,8,17,24,29) have been corrected while some were replaced with new and current references. These were highlighted in pink color in the reference list in the manuscript.

This has been corrected.

Some paragraphs in the discussion have been re-written and some sentences were replaced. These affected paragraphs and sentences were highlighted in pink color in the manuscript.

These comments have been adequately addressed. Some areas of the discussion as mentioned in the text have been re-written. See page 14, line 269, line 272-275.

Page 16, line 324-344,

Page 17, line 349-355.

END#

---

## [Editor Report · Decision Letter 2]

12 Jun 2023

MALARIA INFECTION AND ITS ASSOCIATION WITH SOCIO-DEMOGRAPHICS, LONG LASTING INSECTICIDE NETS USAGE AND HEMATOLOGICAL PARAMETERS AMONG ADOLESCENT PATIENTS IN RURAL SOUTHWESTERN NIGERIA

PONE-D-23-02179R2

Dear Dr. Ibrahim,

Thank you for effecting the necessary corrections as pointed out during the review process. We’re pleased to inform you that your manuscript has been judged scientifically suitable for publication and will be formally accepted for publication once it meets all outstanding technical requirements.

Kind regards,

Segun Isaac OYEDEJI, Ph.D

Academic Editor

PLOS ONE
---

## [Editor Report · Acceptance letter]

6 Jul 2023

PONE-D-23-02179R2 

MALARIA INFECTION AND ITS ASSOCIATION WITH SOCIO-DEMOGRAPHICS, LONG LASTING INSECTICIDE NETS USAGE AND HEMATOLOGICAL PARAMETERS AMONG ADOLESCENT PATIENTS IN RURAL SOUTHWESTERN NIGERIA. 

Dear Dr. Ibrahim:

I'm pleased to inform you that your manuscript has been deemed suitable for publication in PLOS ONE. Congratulations! Your manuscript is now with our production department. 

Kind regards, 

on behalf of

Professor Segun Isaac OYEDEJI 

Academic Editor

PLOS ONE